# Pharmacodynamics and Outcomes of a De-Escalation Strategy with Half-Dose Prasugrel or Ticagrelor in East Asians Patients with Acute Coronary Syndrome: Results from HOPE-TAILOR Trial

**DOI:** 10.3390/jcm10122699

**Published:** 2021-06-18

**Authors:** Cai-De Jin, Moo-Hyun Kim, Kai Song, Xuan Jin, Kwang-Min Lee, Jong-Sung Park, Young-Rak Cho, Sung-Cheol Yun, Michael S. Lee

**Affiliations:** 1Department of Cardiology, Dong-A University Hospital, Busan 49201, Korea; jincaide@zmu.edu.cn (C.-D.J.); songkaihmu@daum.net (K.S.); jinxuan0819@daum.net (X.J.); tnt849@hanmail.net (K.-M.L.); thinkmed@dau.ac.kr (J.-S.P.); nephrone@dau.ac.kr (Y.-R.C.); 2Department of Cardiology, Affiliated Hospital of Zunyi Medical University, Zunyi 563003, China; 3Department of Clinical Epidemiology and Biostatistics, Asan Medical Center, University of Ulsan College of Medicine, Seoul 05505, Korea; ysch97@amc.seoul.kr; 4Division of Cardiology, UCLA Medical Center, Los Angeles, CA 90095, USA; mslee@mednet.ucla.edu

**Keywords:** half-dose reduction, prasugrel, ticagrelor, pharmacodynamics, outcomes, acute coronary syndrome, East Asians

## Abstract

East Asians treated with potent P2Y12 inhibitors (prasugrel or ticagrelor) generally experience more intense platelet inhibitory responses resulting in an increased risk of major bleeding. Whether a half-dose de-escalation strategy improves the net clinical benefit in Korean patients with acute coronary syndrome (ACS) remains uncertain. A total of 120 patients were pragmatically randomized to either prasugrel (*n* = 39, 60 mg loading dose (LD)/10 mg maintenance dose (MD)), ticagrelor (*n* = 40, 180 mg LD/90 mg MD), or clopidogrel (*n* = 41, 600 mg LD/75 mg MD) followed by a half-dose reduction at 1 month, or conventional dose 75 mg clopidogrel. The primary endpoint was the incidence of optimal platelet reactivity (OPR), defined as a P2Y12 reaction unit (PRU) value between 85 and 208 (by VerifyNow) at 3 months. Ticagrelor treatment achieved a significantly lower PRU compared with prasugrel and clopidogrel (31.0 ± 34.5 vs. 93.2 ± 57.1 vs. 153.1 ± 69.4), resulting in the lowest rate of OPR (12.5% vs. 48.7% vs. 63.4%). At 9 months, the minor bleeding was significantly higher with potent P2Y12 inhibitors than with clopidogrel (31.6% vs. 12.2%; HR, 2.93; 95% CI, 1.12–7.75). Only a few patients experienced ischemic complications. In Korean ACS patients, a de-escalation strategy with half-dose ticagrelor and prasugrel from standard dose increased the OPR rate significantly. Half-dose ticagrelor had a lower OPR rate and greater platelet inhibition compared with half-dose prasugrel as well as conventional-dose clopidogrel. Optimal dose reduction strategies for potent P2Y12 inhibitors require further investigation to balance safety and efficacy.

## 1. Introduction

Dual antiplatelet therapy (DAPT) with an oral P2Y12 inhibitor and aspirin remains the cornerstone of antithrombotic therapy in acute coronary syndrome (ACS). The TRITON-TIMI 38 and PLATO trials both reported that a potent oral P2Y12 inhibitor (prasugrel and ticagrelor, respectively) was superior to clopidogrel in reducing ischemic events but came at the cost of increased major bleedings [1,2]. Current American College of Cardiology/American Heart Association and European Society of Cardiology guidelines recommend prasugrel or ticagrelor as the preferred oral P2Y12 inhibitor for the management of ACS patients undergoing percutaneous coronary intervention (PCI) [3]. However, East Asian patients have a different pharmacodynamic and pharmacokinetic response to potent P2Y12 inhibitors compared with non-East Asians. The exposure of active metabolic prasugrel and ticagrelor was higher in Chinese, Japanese, and Korean (~30%) patients compared with Caucasians even after adjusting for bodyweight [4,5]. Compared with clopidogrel, a lower dose of prasugrel (20/3.75 mg) in the PRASFIT-ACS trial [6] provided a favorable risk/benefit profile without increased bleeding, whereas standard-dose ticagrelor in the PHILO and TICAKOREA trials was associated with increased risk of adverse outcomes, including major bleeding, cardiac death, MI, or stroke [7,8]. For low-dose loading and maintenance doses of prasugrel for high platelet reactivity (HPR) and CYP2C19 polymorphic patients, we previously reported that half-dose prasugrel achieved a lower rate of HPR and a higher therapeutic window range (30 days) in Korean PCI patients [9,10]. The optimal dose for potent P2Y12 inhibitors in East Asians with ACS appears to be different with non-East Asians in daily practice.

After acute or subacute stages of PCI in ACS patients, bleeding events are more problematic during the maintenance dose periods [11]. The recent TICO and TWILIGHT studies demonstrated that de-escalation by dropping aspirin at 3 months resulted in favorable outcomes [12,13]. To our knowledge, there is no comparative pharmacodynamics data on dose reduction strategies with potent P2Y12 inhibitors generated to date. We, therefore, sought to evaluate the pharmacodynamic effects and clinical outcomes of a half dose de-escalation strategy of prasugrel and ticagrelor after 1-month standard dosing of potent P2Y12 inhibitors for DAPT treatment in Korean ACS patients after PCI.

## 2. Materials and Methods

### 2.1. Study Design and Population

The design of the HOPE-TAILOR (Half dose of Prasugrel and Ticagrelor in Platelet Response to Acute Coronary Syndrome; URL: https://www.clinicaltrials.gov (accessed on 24 October 2016; Unique identifier: NCT02944123) trial has been previously published [14]. In brief, the study randomized Korean patients (aged between 20 and 75 years) with ACS undergoing PCI at Dong-A University Hospital, Busan, Republic of Korea. Subjects were excluded if they had: (1) a contraindication to aspirin or oral P2Y12 inhibitors (clopidogrel, prasugrel, or ticagrelor); (2) a history of stroke or transient ischemic attack; (3) bodyweight < 60 kg; (4) gastrointestinal bleeding within 6 months, bleeding diathesis, platelet count < 100,000/mm^3^ or hemoglobin < 10 g/dL; (5) hemodynamic unstable or post-cardiopulmonary resuscitation; (6) known severe renal (serum creatinine > 2.5 mg/dL) or hepatic dysfunction (serum liver enzyme or bilirubin > 3 times the normal limit); (7) known severe chronic obstructive pulmonary disease; (8) bradycardia (heart rate <50 beats per minute, 2nd or 3rd-degree atrioventricular block); or (9) concomitant treatment with a CYP3A4 inhibitor or inducer.

The study was originally designed as a prospective, randomized, open-label, blinded, endpoint (PROBE) single-center trial in ACS patients with three different drug loading strategies (prasugrel, ticagrelor, clopidogrel) and de-escalation to half dose with potent P2Y12 inhibitors only at 30 days. However, during the study period, changes to ACC/AHA and ESC guidelines occurred, and prasugrel or ticagrelor became the preferred oral P2Y12 inhibitors over clopidogrel for ACS patients undergoing PCI [3]. We, therefore, modified the study design accordingly. In ST-elevation myocardial infarction (STEMI), patients undergoing primary PCI who were naïve to potent P2Y12 inhibitor were randomized to prasugrel or ticagrelor, whereas patients with a history of PCI on maintenance clopidogrel therapy were reloaded with clopidogrel 600 mg. Non-ST-elevation ACS (NSTE-ACS) patients with ongoing chest pain, ST changes, or troponin elevation followed the original random protocol. However, if there was an absence of ongoing chest pain, or ST changes, or troponin elevation at the initial stage, the patients were loaded with 600 mg clopidogrel. We believe these loading strategies in the HOPE-TAILOR study were pragmatic and reflective of “real-world” settings for ACS patients undergoing PCI (Figure 1). 

All patients were treated with aspirin (100 mg daily) and a standard-dose of P2Y12 inhibitor (prasugrel 60 mg loading dose followed by 10 mg daily, ticagrelor 180 mg loading dose followed by 90 mg twice a day, or clopidogrel 600 mg loading dose followed by 75 mg daily) for 1 month. Following this 1 month period, the patients randomized to ticagrelor had their dose reduced to 45 mg twice daily (with the tablets cut by a knife), while patients randomized to prasugrel had their dose reduced to 5 mg daily (dedicated tablet). Patients randomized to clopidogrel continued with the same maintenance dose. The study protocol was approved by the Institutional Ethics Committee of Dong-A University Hospital. All subjects provided written informed consent prior to enrollment.

### 2.2. Platelet Function Test

Blood samples were collected via antecubital venipuncture into two anticoagulant tubes at 1 and 3 months post-PCI. The platelet function test was performed on an outpatient basis at clinic times between 9:00–12:00 and 13:30–16:30 after confirming administration of the antiplatelet drug administration (Appendix A). Platelet reactivity was measured using the VerifyNow assay (Accumetrics, San Diego, CA, USA). The VerifyNow assay is a whole blood, cartridge-based, optical detection system that measures platelet aggregation [15]. Within the cartridge of the VerifyNow P2Y12 assay is a channel that measures inhibition of the adenosine diphosphate (ADP) P2Y12 receptor. This channel contains ADP as a platelet agonist and prostaglandin E1 as a suppressor of intracellular-free calcium levels to reduce the non-specific contribution of ADP binding to P2Y12 receptors. The numerical results are expressed as P2Y12 reaction units (PRU).

### 2.3. Study Endpoint

The primary endpoint was the incidence of optimal platelet reactivity (OPR), defined as a PRU between 85 and 208 at 3 months post-PCI [16]. High platelet reactivity (HPR) was defined as PRU > 208, while low platelet reactivity (LPR) was defined as PRU < 85 [17]. The safety endpoint was the occurrence of any type of Bleeding Academic Research Consortium (BARC) [18] bleeding at 9 months. The efficacy endpoint was major adverse cardiac and cerebrovascular events (MACCE), defined as the composite of cardiac death, myocardial infarction, target vessel revascularization (TVR) and stroke.

### 2.4. Statistical Analysis

Sample size calculation is based on the primary endpoint to increase PRU values for improving optimal platelet reactivity (OPR) rate at the 3-month time point. In a previous pharmacodynamic study for Korean ACS patients, the OPR rate was 31.7% in standard-dose potent P2Y12 inhibitor (43.6% in prasugrel and 12.5% in ticagrelor) within 2–4 weeks, whereas 60.0% in 5 mg prasugrel [19]. Therefore, we hypothesize that a half-dose of potent P2Y12 inhibitor (5 mg prasugrel and 45 mg ticagrelor) could lead to an 18.3% increase in OPR rate (50%) at 3 months. A total sample size of 136 target subjects will be required for the study to obtain 90% power and α of 0.05, with a possible dropout rate of 10–15%. 

All analyses were performed on an intention-to-treat basis unless stated otherwise. A descriptive analysis was performed by presenting data as mean (standard deviation), median (interquartile range), or number (proportion). Continuous variables were compared with a one-way analysis of variance with post hoc analysis using the Bonferroni method or Kruskal–Wallis test, as appropriate. Categorical variables were compared with χ^2^ statistics or the Fisher exact test. The cumulative incidence for the safety endpoint was generated by Kaplan–Meier estimates, and survival curves were compared using log-rank tests between study groups. The risk of events with potent P2Y12 inhibitors (prasugrel, ticagrelor, or both) relative to clopidogrel treatment was expressed as a hazard ratio (HR) with 95% confidence interval (CI), using a Cox proportional-hazards model. Landmark analysis was performed at 1 month when the dose of prasugrel or ticagrelor was reduced by half, censoring clinical events that occurred before the specified time point. Statistical analyses were performed using IBM SPSS Version 22 (IBM, Chicago, IL, USA). A two-tailed *p* < 0.05 was the criteria for statistical significance. Additional pharmacodynamic comparisons between oral P2Y12 inhibitors (prasugrel vs. ticagrelor vs. clopidogrel) were generated using GraphPad Prism version 7.0.0 for Windows (GraphPad Software Inc., San Diego, CA, USA).

## 3. Results

### 3.1. Baseline Characteristics

From September 2016 through August 2019, a total of 120 ACS patients were randomized to prasugrel (*n* = 39), ticagrelor (*n* = 40), or clopidogrel (*n* = 41) (Figure 1). Patients in the potent P2Y12 inhibitor groups were well matched regarding baseline demographic and clinical characteristics, laboratory data, procedural characteristics, and discharge medication (Table 1). Due to the modified ‘pragmatic’ protocol, the clopidogrel group had less STEMI and more unstable angina patients. In addition, compared to the prasugrel group, the clopidogrel group was older (63 vs. 57 years of age, *p* = 0.048), and mean baseline LDL-C levels were lower than in the ticagrelor group (96.1 vs. 119.7 mg/dL, *p* = 0.029), possibly due to more repeated PCI patients.

### 3.2. Pharmacodynamics Data for Oral P2Y12 Inhibitors 

At 1 month, ticagrelor achieved a significantly lower PRU compared with prasugrel and clopidogrel (20.0 ± 25.3 vs. 40.9 ± 41.4 vs. 159.0 ± 70.1, respectively, *p* < 0.001) and had the lowest rate of OPR (0% vs. 20.5% vs. 56.1%, respectively, *p* < 0.001) (Table 2, Appendix A). Similar results were observed at 3 months, as half-dose ticagrelor achieved a lower PRU compared with half-dose prasugrel and clopidogrel (31.0 ± 34.5 vs. 93.2 ± 57.1 vs. 153.1 ± 69.4, *p* < 0.001) and had the lowest rate of OPR (12.5% vs. 48.7% vs. 63.4%, respectively, *p* < 0.001). De-escalation to half-dose prasugrel at 1 month resulted in an increase in the 3 month PRU (mean differences (95% CI): 52.4 (36.0 to 68.8), *p* < 0.001) (Figure 2A). De-escalation to half-dose ticagrelor at 1 month resulted in an increase in the 3 month PRU (mean differences (95% CI): 11.0 (1.5 to 20.5), *p* = 0.021) (Figure 2B). The PRU did not significantly change with clopidogrel (mean differences (95% CI): −6.1 (−20.6 to 8.5), *p* = 0.402) (Figure 2C).

### 3.3. Clinical Outcomes

#### 3.3.1. Safety Endpoints

At 9 months of clinical follow-up, none of the patients experienced major bleeding (BARC type > 2). The rates of BARC type 1 was higher with ticagrelor and prasugrel compared with clopidogrel (27.5%, 28.2% vs. 12.2%, *p* = 0.057), whereas only three patients who were treated with potent P2Y12 inhibitors experienced BARC type 2 bleeding (one in the prasugrel group and the other two in the ticagrelor group). The composite of BARC type 1 or 2 bleedings occurred in 31.6% of cases with potent P2Y12 inhibitors and 12.2% with clopidogrel (HR, 2.93; 95% CI, 1.12–7.65, log-rank, *p* = 0.021) (Figure 3, Appendix A). Landmark analysis from 1 to 9 months (half-dose periods) demonstrated that potent P2Y12 inhibitors (ticagrelor and prasugrel) had higher rates of BARC type 1 or 2 bleeding compared with clopidogrel (25.7% vs. 7.7%, HR, 3.87; 95% CI, 1.15–13.11; log-rank, *p* = 0.018). However, there was no difference in BARC type 1 or 2 bleeding with ticagrelor and prasugrel (30.8% vs. 20.6%, HR 1.56; 95% CI, 0.62–3.96; log-rank, *p* = 0.341).

#### 3.3.2. Efficacy Endpoints

The overall MACCE rates at 9 months were low (1.7%). One patient (2.4%) who was treated with clopidogrel experienced TVR, while 1 patient (2.5%) treated with ticagrelor experienced a non-fatal myocardial infarction and TVR (Appendix A). 

## 4. Discussion

The HOPE-TAILOR pilot trial is the first head-to-head comparison of pharmacodynamics and clinical outcomes for a de-escalation strategy with half-dose prasugrel versus half-dose ticagrelor with clopidogrel as a reference in East Asian ACS patients. The primary endpoint of OPR incidence at 3 months was lower with half-dose ticagrelor compared with half-dose prasugrel and conventional-dose clopidogrel. The half-dose ticagrelor group had the lowest OPR rate. Half-dosing of potent P2Y12 inhibitors was associated with a higher incidence of minor bleeding at 9 months compared with clopidogrel. None of the patients experienced major bleeding, and ischemic complications were uncommon at 9 months follow-up.

The optimal dose of potent P2Y12 inhibitors in Korean ACS patients is debatable. The PANTASTIC trial was a pharmacodynamic comparison of standard doses of prasugrel versus ticagrelor in Korean patients with STEMI undergoing primary PCI [20]. Both agents provided robust platelet inhibition, resulting in high rates of LPR (approximately 95% at 48 h) after the loading doses were administered. Therefore, a de-escalation strategy that involves reducing the dose of potent P2Y12 inhibitors is reasonable to lower the high rates of LPR in return for lower bleeding risk. A pharmacokinetic and pharmacodynamic study reported that prasugrel 5 mg in East Asians had comparable platelet inhibition compared with prasugrel 10 mg in Caucasians [4]. In the PRASFIT-ACS trial [6], a reduced dose of prasugrel (20 mg loading dose followed by 3.75 mg maintenance dose) in East Asians/Japanese ACS patients provided a favorable risk/benefit profile with preserved ischemia protection without an increase in bleeding, as compared with clopidogrel. The inhibitory effect of prasugrel 2.5 mg maintenance dose by light transmission aggregometry was comparable to clopidogrel 75 mg [21].

Alexopoulos et al. [22] compared the potency of antiplatelet effects between ticagrelor and prasugrel in STEMI patients between acute (24 h) and subacute maintenance phases (5 days). The platelet reactivity was not different at acute phases (2, 4, 6, and 24 h) but was lower in the ticagrelor group than for prasugrel at day 5 (25.6 vs. 50.3, mean PRU value). In the OPTIMA trial, the PRU values for low-dose ticagrelor (60 mg bid) were not significantly different at 1 month with standard-dose ticagrelor (77 ± 41 vs. 59 ± 38) in Korean ACS patients [23]. No patients treated with ticagrelor had HPR, whereas 66.7% of patients treated with clopidogrel had HPR. One of the interesting findings in the current study was that the mean PRU values for ticagrelor and prasugrel at 30 days were well matched with Alexopoulos’s data at 5 days (20.0 vs. 40.9 and 25.6 vs. 50.3 PRU, *p* < 0.05 each) and if it is presumed that Asian patients’ body weights are on average 20% less than Caucasians, the data aligns well (20.0 vs. 40.9 and 20.48 vs. 40.24 PRU, *p* < 0.05 each). Another interesting finding was that the mean PRU value for half-dose ticagrelor at 3 months still had a lower value but was not statistically different to prasugrel 10 mg at 1 month (31.0 ± 34.5 vs. 40.9 ± 41.4 PRU value, *p* = 0.40). Although ticagrelor is a reversible agent and its pharmacodynamic effects are distinct to irreversible drugs like prasugrel or clopidogrel, the standard dose of ticagrelor might be too potent for East Asian patients and could be one explanation for the better result of ISAR-REACT5 [24], as well as negative results in the PHILO and TICAKOREA studies [7,8].

Bleeding events related to potent P2Y12 inhibitors [7,8], or triple therapy (DAPT plus oral anticoagulant) [25], or high bleeding risk (HBR) [26,27], have resulted in adverse outcomes. Previously we reported that the PRECISE-DAPT Score was validated to predict bleeding events in Korean patients taking DAPT after PCI [11]. Additionally, these bleeding events occurred more frequently in the first month after PCI resulting in a hazard ratio of 4.0 (bleedings vs. MACE) [28]. Therefore, early de-escalation within one month represents a potential option to reduce bleeding episodes.

Regarding platelet reactivity, HPR is associated with ischemic events and LPR is associated with bleeding events with hazard ratios of 2.73 and 1.74, respectively [29]. Therefore, OPR might represent a viable option to reduce ischemic and bleeding events, hitting a sweet spot for antiplatelet therapy [30]. In the current study, the reduction of the ticagrelor dose by half to 45 mg after 1 month resulted in the OPR rate increasing from 0 to 12.5% at 3 months. Similarly, reduction of the prasugrel dose by half to 5 mg after 1 month resulted in the OPR rate increasing from 20.5 to 48.7%. Despite the reduction of the doses of these potent P2Y12 inhibitors, no patients exhibited HPR, demonstrating that the antiplatelet effects of these agents were preserved. The half dose of ticagrelor and prasugrel still did not reach an OPR rate in less than half of the patients, especially for ticagrelor in less than 87.5 patients, which might need further attention.

There are several approaches that can be used to downgrade antiplatelet activity. A common way is to switch from a potent P2Y12 inhibitor (prasugrel or ticagrelor) to clopidogrel [31,32,33]. An aspirin-free strategy and shortening of the DAPT duration [34] have also been tested in recent trials. The TICO and TWILIGHT trials reported that standard-dose ticagrelor monotherapy after 3 month DAPT could improve net benefit for individual patients, resulting in a lower risk of major bleeding by approximately 50%, without increasing ischemic risk (cardiac death, MI, or stroke) as compared to ticagrelor and aspirin for 1 year, even in high-risk ACS patients [12,13]. Our study focused on reducing the doses of potent P2Y12 inhibitors after 1 month of full-dose treatment in Korean patients presenting with ACS, although the small sample size was insufficiently powered for broader conclusions.

Regarding potential safety issues, potent P2Y12 inhibitors (ticagrelor and prasugrel) provide greater platelet inhibition and are associated with more frequent BARC type 1 or 2 bleeding compared with clopidogrel at 1month standard maintenance therapy and half doses of these drugs from 1 to 9 months. Because East Asians on ticagrelor or prasugrel treatment are considered to have less favorable net clinical benefits compared with their Caucasian counterparts [30], the HOST-REDUCE-POLYTECH-ACS randomized clinical trial recently provided valuable information on the optimal dose of prasugrel beyond 1 month after PCI, comparing maintenance doses of 5 mg versus 10 mg prasugrel in East Asian ACS patients [35]. In addition, a larger clinical trial involving a tailored dose of ticagrelor in ACS patients undergoing PCI is critically needed. 

We acknowledge several limitations to our findings. The small sample size was not sufficiently powered to evaluate clinical outcomes as the MACE rates were low. In addition, the incidence of dyspnea was not assessed. It remains to be investigated whether ticagrelor dose reductions are related to a decrease in the incidence of dyspnea. Finally, the HOPE-TAILOR was a single-center, open-label, three-armed randomized clinical trial, which was later modified to a pragmatic random design due to updated changes to the guidelines during the enrollment period. Therefore, differences in baseline characteristics could have led to confounding.

## 5. Conclusions

In Korean ACS patients, a de-escalation strategy with half-dose ticagrelor and prasugrel from the standard dose increased the OPR rate significantly. Half-dose ticagrelor was associated with a lower OPR rate and greater platelet inhibition compared to half-dose prasugrel as well as conventional-dose clopidogrel. Optimal dose reduction strategies for potent P2Y12 inhibitors to balance safety and efficacy require further investigation.

## Figures and Tables

**Figure 1 jcm-10-02699-f001:**
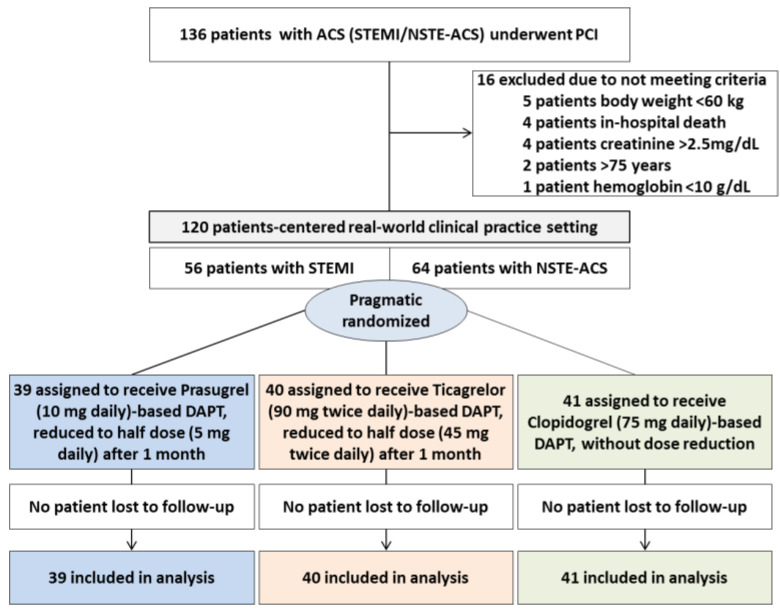
Patient flow and overall clinical trial design.

**Figure 2 jcm-10-02699-f002:**
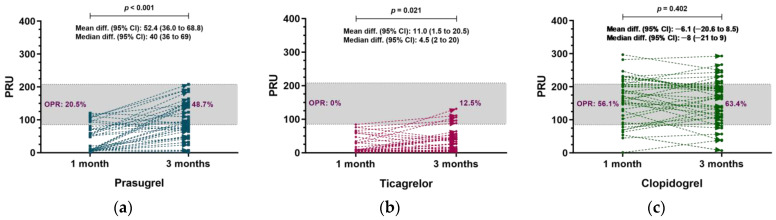
Matched-pairs analysis of 1 month versus 3-month pharmacodynamics in each P2Y12 inhibitor group. (**a**) Prasugrel; (**b**) ticagrelor; (**c**) clopidogrel.

**Figure 3 jcm-10-02699-f003:**
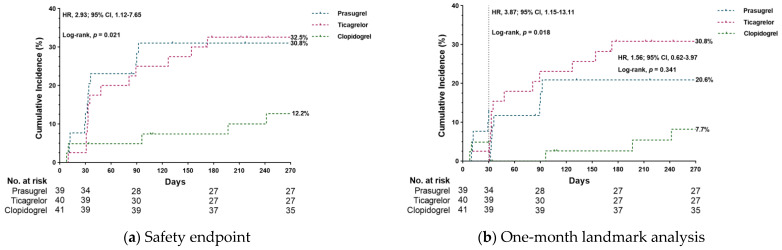
Cumulative incidence of safety endpoints (composite of BARC type 1 or 2 bleedings). (**a**) Kaplan–Meier estimate of safety endpoints at 9 months; (**b**) Landmark analysis at 1 month for safety endpoints.

**Table 1 jcm-10-02699-t001:** Baseline characteristics.

Variable	Overall (*n* = 120)	Prasugrel (*n* = 39)	Ticagrelor (*n* = 40)	Clopidogrel (*n* = 41)	*p*-Value
Age, year	60 ± 10	57 ± 10 ^a^	61 ± 9	63 ± 10 ^b^	0.050
Male gender	109 (90.8)	37 (94.9)	34 (85.0)	38 (92.7)	0.323
Body weight, kg	70.5 ± 9.6	72.1 ± 10.8	69.9 ± 9.2	69.7 ± 9.0	0.543
BMI, kg/m^2^	24.8 ± 2.3	25.0 ± 2.4	24.6 ± 2.3	24.9 ± 2.3	0.802
Diabetes mellitus	29 (24.2)	6 (15.4)	9 (22.5)	14 (34.1)	0.149
Hypertension	52 (43.3)	13 (33.3)	21 (52.5)	18 (43.9)	0.230
Dyslipidemia	22 (18.3)	7 (17.9)	10 (25.0)	5 (12.2)	0.336
Current smoking	27 (22.5)	8 (20.5)	9 (22.5)	10 (24.4)	0.962
Previous MI	14 (11.7)	4 (10.3)	4 (10.0)	6 (14.6)	0.823
Previous PCI	22 (18.3)	5 (12.8)	8 (20.0)	9 (22.0)	0.581
Clinical presentation					0.001
UA	32 (26.7)	6 (15.4)^a^	3 (7.5)^a^	23 (56.1) ^b^	
NSTEMI	32 (26.7)	10 (25.6)	10 (25.0)	12 (29.3)	
STEMI	56 (46.7)	23 (59.0) ^a^	27 (67.5) ^a^	6 (14.6) ^b^	
Hemoglobin, g/dL	13.3 ± 1.9	14.0 ± 2.0	13.5 ± 1.8	13.4 ± 1.8	0.320
Platelet, 10^3^/mm^3^	222.5 ± 53.6	219.6 ± 45.9	233.3 ± 65.3	216.4 ± 46.4	0.501
Troponin I, peak, pg/mL	10.83 (1.10–62.11)	39.74 (4.43–88.31) ^a^	39.27 (6.94–77.19) ^a^	1.13 (0.15–8.07) ^b^	0.001
GFR, mL/min/1.73 m^2^	83.4 ± 24.0	84.2 ± 23.6	84.0 ± 23.4	82.1 ± 25.8	0.945
LDL-C, mg/dL	109.3 ± 38.9	112.0 ± 31.1	119.7 ± 46.1 ^a^	96.1 ± 34.7 ^b^	0.023
Procedural characteristics					
Radial approach	95 (79.2)	34 (87.2)	29 (72.5)	32 (78.0)	0.469
Multivessel disease	47 (39.2)	14 (35.9)	16 (40.0)	17 (41.5)	0.892
Multivessel PCI	33 (27.5)	11 (28.2)	9 (22.5)	13 (31.7)	0.666
PCI					0.844
Stenting	116 (96.7)	38 (97.4)	38 (95.0)	40 (97.6)	
No. > 2	33 (27.5)	9 (23.1)	13 (32.5)	11 (26.8)	0.644
Length (mm)	22 (16–34)	20 (17–27)	26 (18–35)	23 (14–40)	0.289
DCB	2 (1.7)	0 (0)	1 (2.5)	1 (2.4)	
POBA	2 (1.7)	1 (2.6)	1 (2.5)	0 (0)	
Discharge medication					
Aspirin	120 (100.0)	38 (100.0)	40 (100.0)	41 (100.0)	>0.999
ACEi/ARB	30 (25.0)	9 (23.1)	9 (22.5)	12 (29.3)	0.776
Beta-blocker	89 (74.2)	30 (76.9)	30 (75.0)	28 (68.3)	0.661
Statin	113 (94.2)	37 (94.9)	38 (95.0)	38 (92.7)	0.999
Calcium channel blocker	29 (24.4)	9 (23.1)	9 (23.1)	11 (26.8)	0.895
Proton pump inhibitor	25 (20.8)	8 (20.5)	10 (25.0)	7 (17.1)	0.692

Data are presented as mean ± SD, median (IQR), or number (%). ^a^ vs. ^b^, *p* < 0.05. ACEi, angiotensin-converting enzyme inhibitor; ARB, angiotensin receptor blocker; BMI, body mass index; DCB, drug-coated balloon; GFR, glomerular filtration rate; LDL-C, low-density lipoprotein cholesterol; NSTEMI, non-ST-elevation myocardial infarction; PCI, percutaneous coronary intervention; POBA, plain old balloon angioplasty; UA, unstable angina.

**Table 2 jcm-10-02699-t002:** Pharmacodynamics comparison between oral P2Y12 inhibitors.

Time	P2Y12 Reaction Unit (PRU)			Overall Effect	Pairwise Comparisons (95% CI of diff.)		
**1 month**	**Prasugrel**	**Ticagrelor**	**Clopidogrel**	***p* < 0.001**	**P vs. T**	**P vs. C**	**T vs. C**
Mean ± SD	40.9 ± 41.4	20.0 ± 25.3 *	159.0 ± 70.1		−20.9 (−36.2 to −5.6)	118.3 (92.5 to 144.1)	139.2 (113.8 to 162.7)
Median (IQR)	11 (5–72)	7 (3–30) **	167 (97–212)		−4 (−38 to 0)	156 (92 to 152)	160 (122 to 172)
**3** **months**	**Prasugrel**	**Ticagrelor**	**Clopidogrel**	***p* < 0.001**	**P vs. T**	**P vs. C**	**T vs. C**
Mean ± SD	93.2 ± 57.1	31.0 ± 34.5	153.1 ± 69.4		−62.3 (−41.2 to −83.3)	59.9 (31.5 to 88.2)	122.2 (97.8 to 146.5)
Median (IQR)	84 (47–145)	15 (6–45)	169 (107–199)		−69 (−81 to −39)	85 (33 to 95)	154 (101 to 156)

Data are presented as mean ± SD or median (IQR). At 1 month, the PRU value of prasugrel vs. ticagrelor * *p* = 0.008 by *t*-test, ** *p* = 0.023 by Mann–Whitney test. C, clopidogrel; P, prasugrel; T, ticagrelor.

## Data Availability

Data are the property of the authors and can become available by contacting the corresponding author.

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
