# Peer review of "Pharmacodynamics and Outcomes of a De-Escalation Strategy with Half-Dose Prasugrel or Ticagrelor in East Asians Patients with Acute Coronary Syndrome: Results from HOPE-TAILOR Trial"

_jcm, 2021, doi:10.3390/jcm10122699_

Round 1

Reviewer 1 Report

In this study, DE  Jin and colleagies present the data of the HOPE-TAILOR trial. In this trial, a fixed de-escalation strategy by reducing the dose of the P2Y12 inhibitors prasugrel oder ticagrelor 1 month after PCI in ACS, was applied. As primary endpoint, the time in optimal platelet reactivity (OPR) was chosen, secondary endpoints included clinical endpoints. The authors demonstrate that a strategy with low doses of prasugrel oder ticagrelor results in highest rates of OPR.

The study appears well conducted, the manuscript is well written, adequately structed and employs adequate methods.

Major limitations of the study are the relatively small sample size,  the ‘pragmatic’ randomization and the surrogate endpoint of OPR. All of these limitations are adequately discussed.  

Author Response

Response: We appreciate the reviewer for the valuable comments.

Reviewer 2 Report

The authors present a single-center randomized clinical trial (with pragmatic design) of pharmacodynamics regarding three antiplatelet agents and switch to lower dose for the more potent ones.

There are some minor comments as follows:

- Although the pragmatic protocol indicated the use of clopidogrel in patients with previous PCI, there was no significant difference in the respective percentages among the 3 groups. In particular, the percentage in the clopidogrel and ticagrelor group was the same. How is this explained?

- As the authors note, the study is limited by the small sample size, and also the rather short follow-up regarding clinical outcomes and efficacy of treatment. How well does platelet reactivity measured in vitro relate with clinical efficacy?

Author Response

Response 1: Thanks for your comment.

Our patients with previous PCI who were not taking P2Y12 inhibitor were randomized to prasugrel or ticagrelor group, whereas patients with maintenance clopidogrel therapy were reloaded with clopidogrel 600 mg. Therefore, 3 groups were relatively balanced.

Response 2: Thanks for your comment.

The platelet reactivity measured in vitro is a pharmacodynamic response to antiplatelet agents. Low platelet reactivity (LPR) is associated with high bleeding risk, which influences the net clinical benefit. We were interested in LPR and were related with high bleeding events (Choi, SY, Kim MH et al, The challenge for predicting bleeding events by assessing platelet reactivity following coronary stenting, Int J Cardiol, 2016) and VeryfyNow was better test in predicting bleeding events than LTA or MEA (Kim MH et al, Validation of Three Platelet Function Tests for Bleeding Risk Stratification During Dual Antiplatelet Therapy Following Coronary Interventions, Clin Cardiol, 2016). We are planning to start large clinical trial of de-escalation strategy to reduce LPR percentage in PCI patients with potent P2Y12 inhibitor medication.

Reviewer 3 Report

This is a very interesting mechanistic paper suggesting that de-escalation to half dose ticagrelor in Asian patient population could achieve goo OPR and thus no increase of ischemic events or bleeding events. My only question and suggestion to the authors is that all the measurement are done on the background of aspirin while recent studies suggest that perhaps omission of aspirin would address many of the issues. Could you add measurements in patients in whom aspirin was omitted? As authors suggest, this strategy needs further assessment in large trials powered for clinical events.

Author Response

Response: Thanks for your comment.

We did not measure aspirin-free measurements as a protocol (none of the patients omitted the aspirin in the HOPE-TAILOR trial). However, we are going to start a large clinical trial with aspirin omitted protocol after a small pilot trial. We can answer that issue in the near future.

This manuscript is a resubmission of an earlier submission. The following is a list of the peer review reports and author responses from that submission.